# COVseq is a cost-effective workflow for mass-scale SARS-CoV-2 genomic surveillance

Michele Simonetti [1,2,7], Ning Zhang [1,2,3,7], Luuk Harbers [1,2,7], Maria Grazia Milia[4], Silvia Brossa[5], Thi Thu Huong Nguyen [1,2], Francesco Cerutti [4], Enrico Berrino[5,6], Anna Sapino[5,6], Magda Bienko [1,2], Antonino Sottile[5], Valeria Ghisetti [4✉] & Nicola Crosetto [1,2✉]

While mass-scale vaccination campaigns are ongoing worldwide, genomic surveillance of severe acute respiratory syndrome coronavirus 2 (SARS-CoV-2) is critical to monitor the emergence and global spread of viral variants of concern (VOC). Here, we present a streamlined workflow—COVseq—which can be used to generate highly multiplexed sequencing libraries compatible with Illumina platforms from hundreds of SARS-CoV-2 samples in parallel, in a rapid and cost-effective manner. We benchmark COVseq against a standard library preparation method (NEBNext) on 29 SARS-CoV-2 positive samples, reaching 95.4% of concordance between single-nucleotide variants detected by both methods. Application of COVseq to 245 additional SARS-CoV-2 positive samples demonstrates the ability of the method to reliably detect emergent VOC as well as its compatibility with downstream phylogenetic analyses. A cost analysis shows that COVseq could be used to sequence thousands of samples at less than 15 USD per sample, including library preparation and sequencing costs. We conclude that COVseq is a versatile and scalable method that is immediately applicable for SARS-CoV-2 genomic surveillance and easily adaptable to other pathogens such as influenza viruses.

[1] Bienko-Crosetto Lab for Quantitative Genome Biology, Division of Genome Biology, Department of Medical Biochemistry and Biophysics, Karolinska Institutet, Stockholm, Sweden. [2] Science for Life Laboratory, Solna, Sweden. [3] Department of Breast Surgery, Qilu hospital of Shandong University, Ji'nan, China. [4] Laboratory of Microbiology and Virology, Ospedale 'Amedeo di Savoia', Turin, Italy. [5] Instituto di Candiolo FPO—IRCCS, Candiolo, Turin, Italy. [6] Department of Medical Sciences, University of Turin, Turin, Italy. [7] These authors contributed equally: Michele Simonetti, Ning Zhang, Luuk Harbers. ✉email: valeria.ghisetti@gmail.com; nicola.crosetto@ki.se

Since the identification of Severe Acute Respiratory Syndrome Coronavirus 2 (SARS-CoV-2) as the causative agent of coronavirus disease 2019 (COVID-19)[1], thousands of SARS-CoV-2 genomes have been sequenced worldwide and the sequences have been made publicly available on the global initiative on sharing all influenza data (GISAID)[2] (https://www.gisaid.org/). This has enabled a phylogenetic reconstruction of the viral spread and evolution across different countries and continents at an unprecedented scale[3], allowing the rapid identification of new viral variants, including the UK[4], South African[5], and Brazilian[6] variants of concern (VOC), which have now been reclassified by the World Health Organization (WHO) as Alpha, Beta, and Gamma, respectively. SARS-CoV-2 whole-genome sequencing (WGS) based on next-generation sequencing (NGS) has been used for genomic surveillance to track infections in hospitals and community settings[7,8], as well as to monitor viral outbreaks in breeding farms, such as mink farms[9]. More recently, with the advent of mass-scale vaccination campaigns worldwide, SARS-CoV-2 genomic surveillance has become vital to rapidly trace the emergence and spread of VOC with potentially reduced susceptibility to the current vaccines[10,11]. In this context, the availability of scalable and cost-effective methods for centralized sequencing of hundreds or potentially thousands of samples per week would be greatly beneficial.

Various NGS-based approaches have been developed to perform SARS-CoV-2 WGS using different sequencing platforms. These include direct RNA sequencing and metagenomics[12–14], amplicon-based methods[13,15,16], and oligonucleotide capture-based methods[13,16–18]. Recently, numerous commercial kits based on the above methods have become available on the market and are being deployed in SARS-CoV-2 genomic surveillance[19]. However, existing commercial solutions are very costly and/or difficult to scale up largely because, typically, one sequencing library must be prepared for each individual sample and multiple indexed libraries must be carefully quantified and balanced before pooling them together prior to sequencing. This limits the number of samples that can be sequenced on a weekly basis, increasing the risk of missing emerging variants of potential concern. In addition to its applications for WGS, NGS has also been used for mass-scale SARS-CoV-2 testing, such as in the SwabSeq method[20]; however, the latter does not provide full genome coverage. To counteract these limitations, here we present a versatile, scalable, and cost-effective workflow—COVseq—that can be used to prepare multiplexed WGS libraries from many SARS-CoV-2 samples in parallel. We validate COVseq using RNA extracted from a SARS-CoV-2 viral culture as well as 274 diagnostic samples collected in three different phases of the ongoing pandemic at two hospitals in Italy. We demonstrate that the genome sequences obtained by COVseq are compatible with downstream phylogenomic analyses, including detailed phylogenetic reconstruction of a COVID-19 nosocomial outbreak that occurred in January 2021 at a single hospital in Italy. Lastly, we perform a real-life cost analysis based on our experience with the genomic surveillance program that we recently initiated for the Piemonte Region in North-West Italy, demonstrating that COVseq is a highly cost-effective method for SARS-CoV-2 genomic surveillance, including in low-income countries.

## Results

**COVseq enables near-complete coverage of the SARS-CoV-2 genome**. The CUTseq method, which we previously described[21], enables a cost-effective preparation of highly multiplexed DNA sequencing libraries, by using restriction enzymes to barcode multiple samples before pooling them together into a single library. When the SARS-CoV-2 pandemic started, we sought to adapt CUTseq to sequence many SARS-CoV-2 genomes in parallel at an affordable cost. To this end, we designed a workflow—COVseq—that begins with a multiplexed PCR assay developed by the U.S. Centers for Disease Control and Prevention (CDC) to amplify the whole SARS-CoV-2 genome, followed by CUTseq on the resulting purified amplicons (Fig. 1a, b, Supplementary Data 1, 2, and "Methods"). A step-by-step COVseq protocol is available in Supplementary Methods and at Protocol Exchange[22]. We first determined the breadth of coverage of the SARS-CoV-2 genome that could theoretically be achieved using two frequently cutting restriction enzymes compatible with CUTseq[21]. In silico simultaneous digestion with two 4-base cutters, MseI and NlaIII, predicted that 96.1% of the SARS-CoV-2 genome and 100% of the S region that is targeted by all current vaccines would be covered by 150 nucleotides (nt) single-end (SE) sequencing, while 98.8% of the whole genome and 100% of the S region would be covered by SE300 sequencing (Table 1 and "Methods"). In the latter scenario, only one out of 32 single-nucleotide variants (SNVs) in the UK, South African, and Brazilian VOC falls into a so-called "dark region" and thus would not be covered by COVseq (Supplementary Fig. 1a). We then tested the COVseq workflow by preparing a single library from the RNA extracted from the supernatant of a SARS-CoV-2 culture, using MseI and NlaIII either alone or in combination to digest the pre-amplified SARS-CoV-2 genome ("Methods"). In line with our theoretical expectations, double digestion and SE150 sequencing resulted in ~95% of the whole SARS-CoV-2 genome and S region covered at 10× sequencing depth and ~99% covered at 1× (Fig. 1c, d and Supplementary Data 3).

Next, we sought to apply COVseq to left-over SARS-CoV-2-positive RNA samples extracted from nasopharyngeal swabs taken for routine SARS-CoV-2 testing. We prepared a single COVseq library from 29 SARS-CoV-2-positive samples collected during Phase 1 of the pandemic at a hospital in Turin, Italy (OAS-29 samples in Supplementary Data 4 and "Methods"). To minimize the volume of reagents used and hence the cost per sample, we performed all the barcoding reactions in nanoliter volumes, by leveraging on a contactless nanodispensing device (I-DOT One), which we previously used for high-throughput CUTseq[21] ("Methods"). As a reference, we prepared individual sequencing libraries from each of the 29 samples using a commercial kit (NEBNext), which is compatible with SARS-CoV-2 WGS (Supplementary Fig. 1b, c, Supplementary Data 3, and "Methods"). The number of reads per sample inversely scaled with the corresponding cycle threshold (Ct) value in both COVseq and NEBNext and was highly correlated between them (Pearson's correlation coefficient, PCC: 0.87) (Fig. 1e, f and Supplementary Fig. 2a). Furthermore, the breadth of SARS-CoV-2 genome coverage at 10× was highly correlated (PCC: 0.95) between the two methods, independently of the sequencing platform and modality (NextSeq 500 PE150 vs. MiSeq PE300) (Fig. 1g and Supplementary Fig. 2b). Of note, in samples with high Ct value (> 35), a sizable fraction of the reads was aligned to the human genome, independently of the method used to prepare the libraries (Fig. 1h). To further validate COVseq, we assessed how many SNVs identified by NEBNext in 20 samples with low Ct (≤ 35)—a threshold that allows reliable SNV calling—were also detected by COVseq (Supplementary Data 5 and "Methods"). The number of SNVs per sample was highly correlated (PCC: 0.99) between COVseq and NEBNext, and 205 out of 217 SNVs (95.4%) were detected by both methods (Fig. 1i, j and Supplementary Fig. 2c). The remaining SNVs that were detected by only one method had a substantially lower depth of coverage (Supplementary Fig. 2d). Importantly, most of the genomic locations of the SNVs in the UK, South African, and Brazilian VOC were covered by COVseq at a depth (≥15 reads) that would be sufficient to identify them, if they were present in these samples (Fig. 1k).

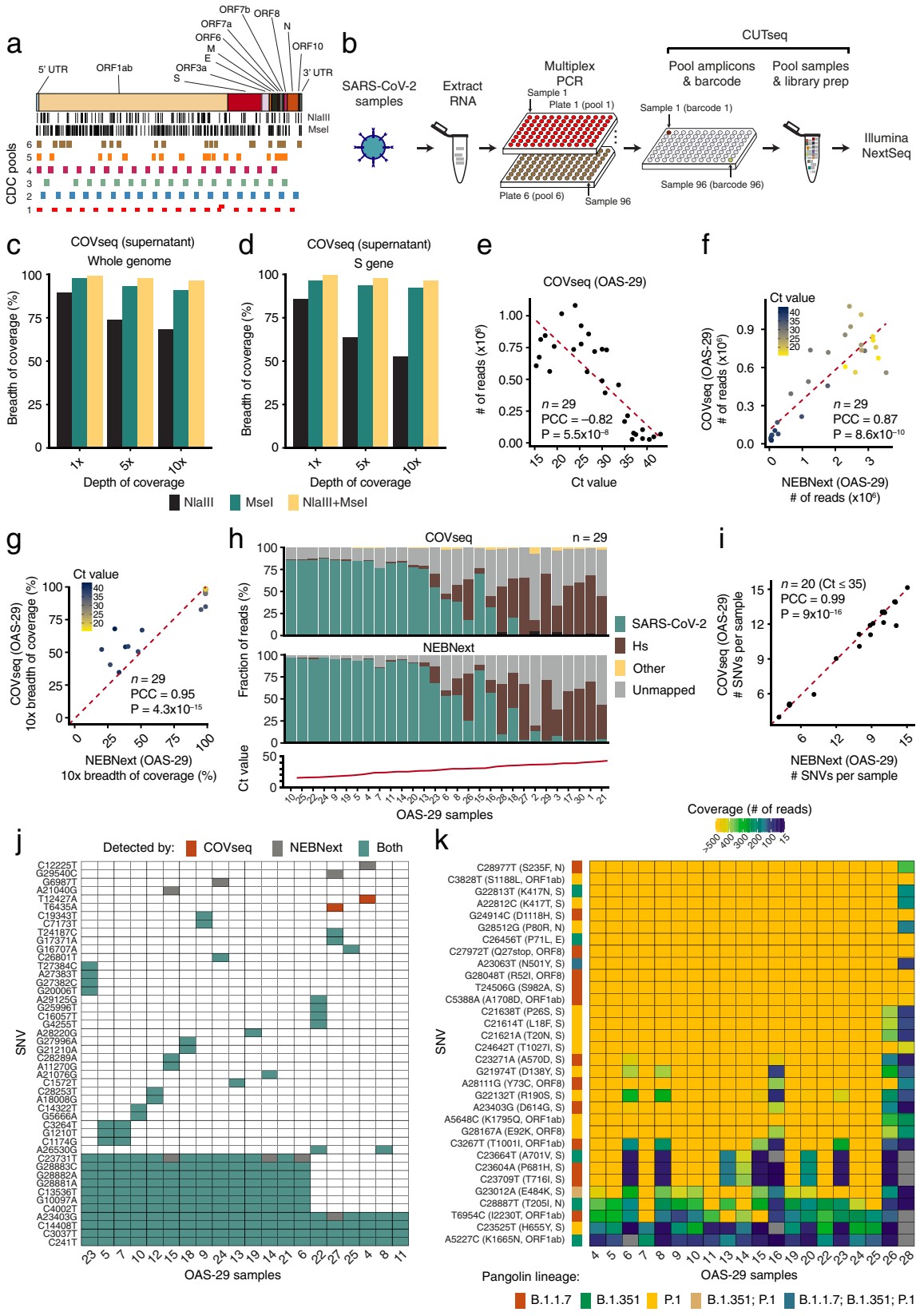

**COVseq can reproducibly identify SNVs including those found in emerging VOC.** Having demonstrated the feasibility and analytical validity of COVseq, we sought to assess the reproducibility and sensitivity of our method in detecting known VOC. To this end, we prepared three replicate (Rep) COVseq libraries, each

containing 95 samples (samples OAS-95 in Supplementary Data 4), including 7 samples suspected to contain the UK variant based on PCR testing (S-gene target failure in Thermo Fisher TaqPath Reverse Transcription (RT) PCR assay) (Supplementary Data 3 and "Methods"). Remarkably, the breadth of SARS-CoV-2 genome

**Fig. 1 COVseq implementation and validation. a** Location along the Severe Acute Respiratory Syndrome Coronavirus 2 (SARS-CoV-2) genome (top) of MseI and NlaIII recognition sites (vertical black bars) and Centers for Disease Control and Prevention (CDC) multiplexed PCR assay amplicon pools (colored rectangles). Gene names (top) are according to the reference SARS-CoV-2 sequence NC_045512.2. **b** Schematic high-throughput COVseq workflow. Purified RNA samples (e.g., extracted from nasal- or oro-pharyngeal swabs) are first equally distributed in corresponding wells of six 96-well plates and amplified using six different PCR primer pools (one pool per plate) to amplify the amplicons shown in (**a**). After PCR, the contents of the wells in the six 96-well plates are pooled into the corresponding wells of a new 96-well plate and purified. Afterwards, 96 CUTseq adapters (see Supplementary Data 2) are used to barcode each sample individually, before all the samples are pooled together into the same sequencing library. Alternatively, 384 samples can be barcoded separately before being pooled together, by using the 384 CUTseq adapters listed in Supplementary Data 2. **c** Percentage of bases in the SARS-CoV-2 reference genome covered by COVseq at varying sequencing depths (SE150 sequencing) for three different libraries prepared from RNA extracted from the supernatant of a viral culture, using genome digestion with one or two restriction enzymes (MseI and NlaIII). **d** Same as in (**c**), but for the S gene encoding the spike protein. **e** Inverse correlation between the cycle threshold (Ct) determined by RT-PCR and the number of reads, for OAS-29 samples (see Supplementary Data 4) sequenced by COVseq (MiSeq PE300). **f** Correlation between the total number of reads obtained with COVseq vs. NEBNext for the same samples in (**e**). **g** Correlation between the breadth of coverage at 10× sequencing depth obtained by COVseq vs. NEBNext for the same samples in (**e**). **h** Percentage of sequencing reads aligned to the SARS-CoV-2 reference genome, human reference genome (Hs), other genomes or unmapped, for the same samples in (**e**). The bottom plot shows the Ct value of each sample. **i** Correlation between the number of single-nucleotide variants (SNVs) per sample detected by COVseq (PE300) vs. NEBNext (SE75) in 20 (*n*) out of 29 OAS-29 samples with Ct ≤ 35. **j** Matrix showing the SNVs detected by COVseq, NEBNext, or both in the 20 OAS-29 samples with Ct ≤ 35. **k** Heatmap of the depth of coverage at the genomic positions of all the SNVs defining the UK (B.1.1.7), South African (B.1.351) and Brazilian (P.1) variants of concern (VOC) for the 20 OAS-29 samples with Ct ≤ 35 sequenced by COVseq. Gray color indicates locations that would have insufficient coverage to call SNVs (< 15 reads). In brackets: amino acid change and SARS-CoV-2 gene affected. In (**e–g**) and (**i**): each dot represents a sample; *n* number of samples, PCC Pearson's correlation coefficient, *P*, *t*-test, two-tailed. In (**e**) and (**f**), the dashed red line represents the linear regression fit. In (**g**) and (**i**), the dashed red line is the bisector. In (**f**) and (**g**), each sample is color-coded based on the corresponding Ct value. For sample IDs in (**h**) and (**j, k**), see Supplementary Data 4. OAS Ospedale Amedeo di Savoia.

**Table 1 Theoretical fraction (%) of the SARS-CoV-2 genome covered by CUTseq using one or two restriction enzymes (MseI and NlaIII) and different sequencing read lengths (nt).**

| Read length | Region | NlaIII | MseI | NlaIII + MseI |
|---|---|---|---|---|
| 150 | Whole genome | 65.8% | 90.0% | 96.1% |
| 300 | Whole genome | 84.4% | 96.5% | 98.8% |
| 150 | S | 44.7% | 96.0% | 100% |
| 300 | S | 67.4% | 100% | 100% |

coverage at 10× was close to 100% even in samples with high Ct (> 35) and was highly correlated (PCC > 0.98) between replicate samples (Fig. 2a, b and Supplementary Fig. 3a–c). Accordingly, the number of SNVs per sample was highly correlated (PCC > 0.96) between replicates and most of the SNVs identified were detected in all three replicates (Fig. 2c–e, Supplementary Fig. 3d–f, and Supplementary Data 5). Notably, all the seven samples with S-gene target failure suspected to contain the UK variant were correctly assigned to the B.1.1.7 Pangolin lineage based on COVseq (Fig. 2e and Supplementary Data 5). To further highlight the reproducibility of COVseq, we sequenced four additional replicate libraries prepared using an additional set of 55 samples from a different hospital in Italy (samples CCI-55 in Supplementary Data 4), which confirmed the ability of our method to detect SNVs in a highly reproducible manner (Supplementary Fig. 4a–c, Supplementary Data 3, 5, and "Methods"). These results demonstrate that COVseq can reproducibly identify SARS-CoV-2 genomic variants, including emerging VOC.

**COVseq data are compatible with phylogenetic reconstruction of COVID-19 clusters.** Next, we explored whether the sequencing data generated by COVseq are compatible with downstream phylogenomic analyses. To this end, we used Nextstrain[23] and a random selection of SARS-CoV-2 genomes from GISAID[2] in order to build phylogenetic trees visualizing the hierarchical distribution of the 179 samples that we sequenced with COVseq

("Methods"). As expected, these samples formed several distinct clusters in different branches of the phylogenetic tree, reflecting their time and location of collection (Fig. 3a, b). The seven OAS-95 samples, which were assigned to the B.1.1.7 (UK) variant based on COVseq, clustered in the 20I operational taxonomic unit clade together with all the other GISAID samples that were classified as UK variant (Fig. 3a, b). Notably, 87 of the 95 OAS-95 samples, which were collected in Jan 2021 during a nosocomial COVID-19 outbreak at a hospital in Turin, Italy, formed a clearly distinct cluster (Fig. 3a, b). Closer inspection of this cluster suggested that the outbreak had most likely originated in one ward (internal medicine, case index #1) and then spread to two other wards (orthopedics and cardiology) (Fig. 3c). These results demonstrate that COVseq generates high-quality genome sequences that are compatible with downstream phylogenomic analyses, including phylogenetic reconstructions of COVID-19 clusters.

**COVseq is suitable for mass-scale SARS-CoV-2 genomic surveillance.** Last, we performed a comparative cost analysis to assess the applicability of COVseq in mass-scale genomic surveillance programs. Considering only library preparation costs and assuming to pool 96 samples into the same COVseq library, we determined that the cost per sample would be ~$37, $22, and $20 for 1000, 10,000, and 100,000 samples (Fig. 4a–c and Supplementary Notes). In contrast, using the commercial kits that we considered for comparison (CleanPlex, NEBNext, and Nextera) would result in an average cost per sample up to five times higher (Fig. 4a–d and Supplementary Notes). A further decrease in the COVseq cost per sample (as well as in the hands-on time) could be achieved by substituting the CDC multiplex PCR with the one developed by the ARTIC network (https://artic.network/), which only requires two amplicon pools (Supplementary Fig. 5a and "Methods"). Using this workflow, 100,000 samples could be processed by COVseq and sequenced on Illumina's NovaSeq 6000 at a cost of ~$14 per sample (Supplementary Notes). Notably, we are now testing ARTIC-COVseq in the frame of a SARS-CoV-2 genomic surveillance program that we are conducting for the Piemonte Region in Italy (Supplementary Fig. 5b). Collectively, these results indicate that COVseq is a sensitive, reproducible,

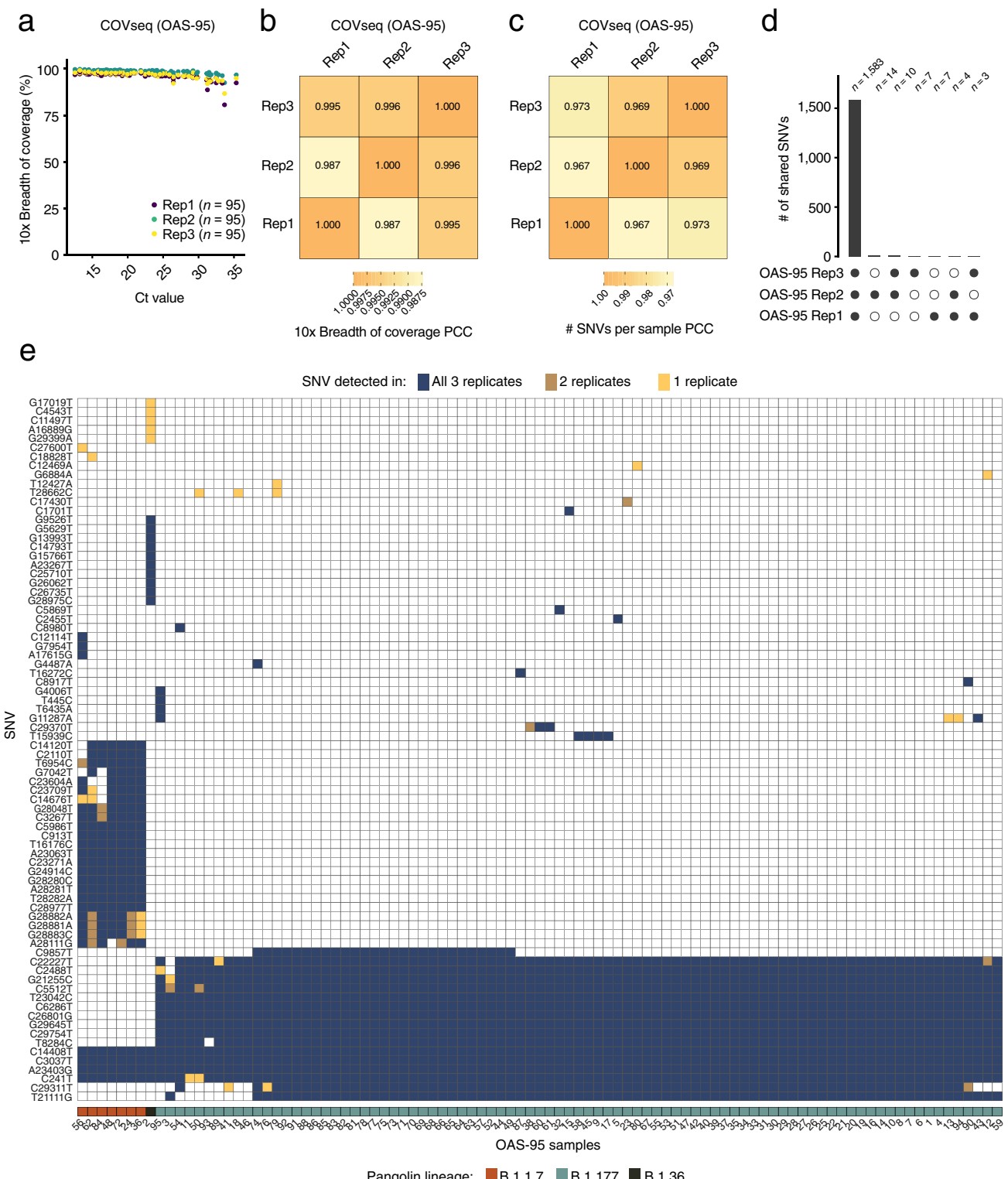

**Fig. 2 COVseq reproducibility. a** Breadth of coverage at 10× sequencing depth for three replicate (Rep) COVseq libraries each including 95 samples (samples OAS-95 in Supplementary Data 4). **b** Correlation matrix showing the Pearson's correlation coefficient (PCC) of the breadth of coverage at 10× sequencing depth between the three replicate (Rep) libraries shown in (**a**). **c** Same as in (**b**), but for the number of single-nucleotide variants (SNVs) in each of the OAS-95 samples. **d** Bar plot showing the number (n) of SNVs shared by two or three of the Rep libraries in (**a**). **e** Matrix showing the list of all SNVs detected in the OAS-95 samples, in one, two, or all of the Rep libraries. The samples were ranked based on their similarity. The Pangolin lineage assigned to each sample is shown at the bottom. For sample IDs, see Supplementary Data 4. OAS Ospedale Amedeo di Savoia.

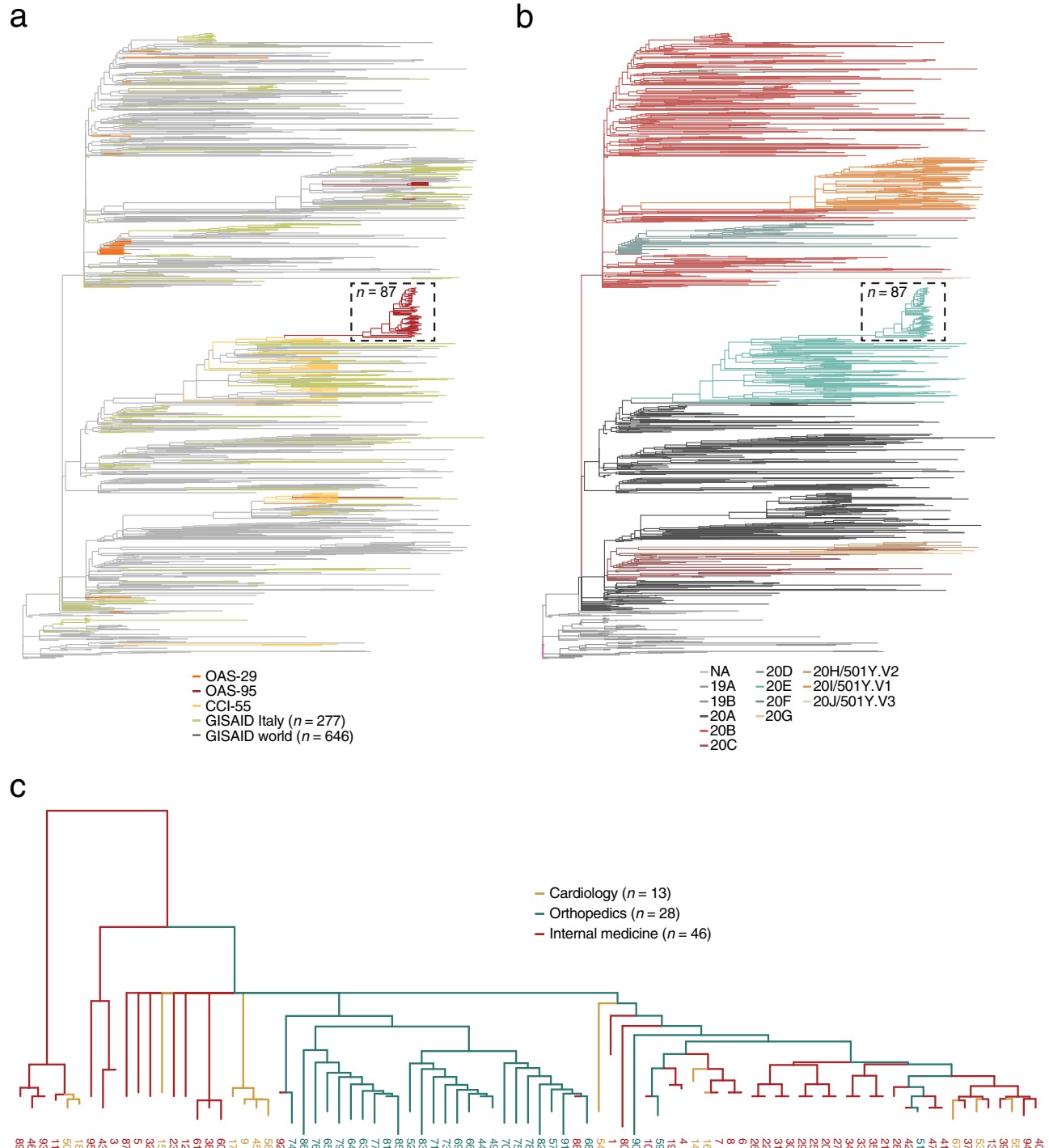

**Fig. 3 Phylogenetic analyses using COVseq data. a** Newick trees showing the phylogeny of the 179 samples sequenced by COVseq together with 909 randomly selected SARS-CoV-2 sequences downloaded from the global initiative on sharing of influenza data (GISAID)[2], including 277 sequences from Italy and 646 from the rest of the world. Colors indicate the geographical origin of the samples. Dashed rectangle: cluster of 87 (*n*) cases from a nosocomial outbreak that occurred in January 2021 at a hospital in Turin, Italy and involved three different wards (orthopedics, cardiology, and internal medicine). **b** Same tree as in (**a**), but with colors indicating the operational taxonomic unit (OTU) clades. Abbreviations refer to the different clades. NA not assigned. **c** Magnified view of the cluster encircled by the dashed rectangle in (**a**) and (**b**). *n* number of samples.

scalable, and highly cost-effective method that is suitable for mass-scale SARS-CoV-2 genomic surveillance.

## Discussion
As the COVID-19 pandemic continues to rage worldwide, the use of genomic surveillance to monitor SARS-CoV-2 outbreaks in communities, healthcare settings as well as in farms where thousands of susceptible animals live in close proximity has become increasingly important[7–9]. Moreover, with the emergence of new viral variants with potentially higher infectivity and/or pathogenicity, there is a cogent need for streamlined and cost-effective approaches that could be deployed for sequencing thousands of viral samples per week. This is particularly relevant in the context of the ongoing mass-scale vaccination campaigns,

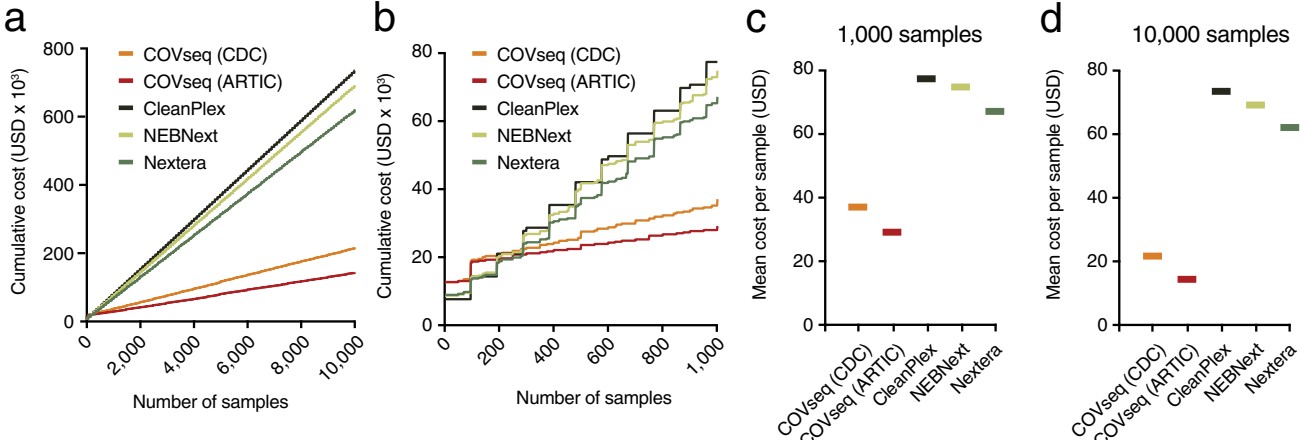

**Fig. 4 COVseq applicability for SARS-CoV-2 genomic surveillance. a** Cumulative reagent cost curves for preparing sequencing libraries from up to 10,000 samples by COVseq using the CDC (CDC-COVseq) or ARTIC (ARTIC-COVseq) multiplexed PCR strategy vs. three different commercial kits (CleanPlex, NEBNext, and Nextera). CDC Centers for Disease Control and Prevention. **b** Same as in (**a**), but for up to 1000 samples. **c** Average cost per sample based on the final cumulative cost and total number of samples shown in (**a**) and (**b**). See Supplementary Notes for a detailed description of how the cost analysis was performed. **d** Same as in (**c**), but for up to 10,000 samples.

in order to quickly detect and respond to the possible emergence of variants conferring resistance to the existing vaccines[10,11]. The main bottleneck toward this goal is that existing commercial solutions for preparing sequencing libraries from thousands of SARS-CoV-2 samples are costly and time-consuming, mainly because the reagent volumes used are high (microliter range) and because, typically, a single library must be prepared from each sample and quantified before sequencing. In contrast, the COVseq method that we have described here allows constructing highly multiplexed sequencing libraries starting from small volumes of purified RNA samples, and it only requires a nano-dispensing device to drastically reduce reagent volumes and therefore the cost per sample. The I-DOT nanodispensing device that we have used here is a versatile bench-top instrument that requires minimal maintenance and training, while drastically reducing plastic consumption by operating in contactless mode. Our cost analysis and ongoing experience in the Piemonte Region in Italy indicates that COVseq is a highly cost-effective approach that could be readily and widely adopted for genomic surveillance of the ongoing pandemic, even in low-income countries. In this context, COVseq is complementary to other sequencing-based methods for mass-scale SARS-CoV-2 testing, which do not provide whole-genome sequence information, such as SwabSeq[20].

Although COVseq is designed to achieve SARS-CoV-2 WGS, the fact that the primers in the CDC or ARTIC pools can be ordered separately allows, in principle, to sequence only a fraction of the SARS-CoV-2 genome, therefore increasing the number of samples that can be sequenced in parallel for the same total cost. This could be applied, for example, to sequence the S gene—which encodes the Spike protein that is targeted by all existing SARS-CoV-2 vaccines—in a much larger number of samples than would actually be possible by WGS, in order to promptly detect the emergence of new amino acid changing variants potentially impacting viral transmission and/or vaccine efficacy.

Since COVseq relies on restriction enzymes that cut the genome non-randomly, some parts of the SARS-CoV-2 genome might not be covered at a depth sufficiently high to reliably call mutations in these regions. However, our data demonstrate that a combination of two restriction enzymes (MseI and NlaIII) and sufficiently long sequencing reads (PE150 or PE300) results in near-complete (98.8%) SARS-CoV-2 genome coverage and allows detecting the recently emerged VOC.

Although COVseq is tailored for SARS-CoV-2, it can be easily adapted to other RNA viruses, such as influenza viruses, as well as to DNA viruses. Indeed, a quick survey of existing multiplexed PCR assays for viruses other than SARS-CoV-2 shows that CUTseq could be readily implemented to sequence the genome of Influenza A and B viruses as well as Dengue, using the same restriction enzymes as in COVseq (Supplementary Fig. 6 and Supplementary Table 1). Different enzyme combinations could also be tested to achieve optimal genome coverage, depending on the pathogen of interest. In conclusion, we envision that COVseq will play an important role in the genomic surveillance of the ongoing and future pandemics.

## Methods

**Samples**. To test the feasibility of COVseq, we used RNA extracted from the supernatant of a SARS-CoV-2 human viral culture on Vero E6 cells, previously established at the Ospedale Amedeo di Savoia (OAS) hospital in Turin, Italy. In addition, to technically validate our method, we used 274 fully anonymous, left-over SARS-CoV-2-positive RNA samples that were collected during Phases 1 (March to April 2020), 2 (October to November 2020), and 3 (February to March 2021) of the pandemic at OAS and Candiolo Cancer Institute (CCI) in Turin, Italy, respectively. The RNA samples were extracted using the EasyMag extraction kit (Biomérieux, cat. no. 280133-280134-200292-280130-280131-280132-280135-280146) on the NUCLISENS easyMAG instrument (Biomérieux) (samples OAS-29 in Supplementary Data 4); the Viral RNA Isolation Kit (Liferiver, cat.no. ME-0044/ME-0045) (samples OAS-95 in Supplementary Data 4); or the MagMax Viral/Pathogen Nucleic Acid Extraction kit (Applied Biosystems, cat. no. A42352) on the KingFisher instrument (Thermo Fisher Scientific) (samples CCI-55 in Supplementary Data 4) or on the Elitech InGenius instrument (ELITechGroup) (samples CCI-95 in Supplementary Data 4). In all cases, there was no DNase treatment step in the RNA extraction procedure. The samples encompassed a broad range of Ct values based on real-time PCR (see Supplementary Data 4). Since the study was conducted on anonymous left-over samples and no clinical and personal information was collected, no informed consent subscription was required. The study was approved by the Ethical Committee of the CCI (permit no. 57/2021) and by the Swedish Ethical Review Authority (permit no. 2020-06694).

**Real-time PCR**. The differences below reflect the fact that each institution providing the samples adopted different approved diagnostic kits in different phases of the pandemic.

*Supernatant and OAS-29 samples*. We performed SARS-CoV-2 real-time PCR on these samples using the Liferiver Novel Coronavirus (2019-nCoV) Real-Time Multiplex RT-PCR kit (Shanghai ZJ Bio-Tech CO. ltd. Liferiver, cat. no. RR-0479-02-ZJ) following the manufacturer's instructions. The kit allows simultaneous detection of three genes: ORF1ab (RdRP), N, and E. For each sample, we prepared a 25-μL reaction containing 5 μL of purified RNA and 20 μL of PCR master mix. PCR conditions were as follows: (i) 45 °C for 10 min; (ii) 95 °C for 3 min; (iii) 45

cycles of 95 °C for 15 s and 58 °C for 30 s. Ct ≤ 43 was set as a cutoff for SARS-CoV-2 positivity.

*CCI-55 and OAS-95 samples*. We performed SARS-CoV-2 real-time PCR on these samples using the TaqPath COVID-19 CE-IVD RT-PCR kit (Thermo Fisher Applied Biosystems, cat.no. A48067) following the manufacturer's instructions for RNA samples extracted from up to 200 µL of input material. The kit allows simultaneous detection of three genes: ORF1ab, N, and S. Ct ≤ 37 was set as cutoff for SARS-CoV-2 positivity.

*CCI-95 samples*. We performed SARS-CoV-2 real-time PCR on these samples using the TaqPath COVID-19 CE-IVD RT-PCR kit (Thermo Fisher Applied Biosystems, cat.no. A48067) or the SARS-CoV-2 ELITe MGB kit (ELITechGroup, cat. no. RTS170ING) following the manufacturer's instructions.

**RT and multiplexed PCR**. To be able to sequence the SARS-CoV-2 genome even in high Ct value samples containing only small amounts of SARS-CoV-2 RNA, we adopted a SARS-CoV-2 multiplexed PCR protocol (v200325.2) developed by the U. S. Centers for Disease Prevention and Control (CDC) (https://github.com/CDCgov/SARS-CoV-2_Sequencing/tree/master/protocols). For CCI-95 samples (see Supplementary Data 4), we adapted the multiplexed PCR protocol (v3) developed by the ARTIC network (https://www.protocols.io/view/ncov-2019-sequencing-protocol-v3-locost-bh42j8ye). We performed all the following steps in a biosafety level 2 (BSL-2) lab using standard reagent volumes. Briefly, we first reversed transcribed each RNA sample, by preparing a mix containing 5 µL of purified RNA, 1 µL of 50 µM random hexamers (Thermo Fisher Scientific, cat. no. N80800127), 1 µL of 10 mM dNTPs (Thermo Fisher Scientific, cat. no. R0191), and 6 µL of Nuclease-Free Water (Thermo Fisher Scientific, cat. no. AM9932) and incubating the reaction for 5 min at 65 °C, after which we cooled the same on ice. To generate single-stranded cDNA, we added (in order) 4 µL of SuperScript IV buffer (Thermo Fisher Scientific, cat. no. 18090050), 1 µL of 0.1 M DTT (Thermo Fisher Scientific, cat. no. 18090050), 1 µL of RNase OUT (Thermo Fisher Scientific, cat. no. 10777-019), and 1 µL of SSIV reverse-transcriptase enzyme (Thermo Fisher Scientific, cat. no. 18090050) to the RT mix and incubated it in a thermocycler using the following program: 23 °C 10 min, 50 °C for 10 min, 85 °C for 10 min and hold at 4 °C. Afterwards, we added 1 µL of RNASe H (Thermo Fisher Scientific, cat. no. 18021071) to the sample and incubated it for 20 min at 37 °C. This step is optional and was not performed when using the ARTICv3 protocol. For multiplexed PCR using the CDC approach, we first mixed equal volumes of the corresponding forward (F) and reverse (R) primers (Integrated DNA Technologies) diluted at 50 µM in Nuclease-Free Water (see Supplementary Data 1 for all primer sequences). We then prepared six primer pools according to the aforementioned CDC protocol, by mixing an equal volume of each F + R primer pair in a pool (see Supplementary Data 1 for the primer pairs contained in each pool). To perform the multiplexed PCR, for each primer pool, we aliquoted 3 µL of each cDNA prepared as described above in six separate PCR tubes prefilled with the following reaction mix: 15 µL of NEBNext Q5 Hot Start HiFi PCR Master Mix (NEB, cat. no. M0543L), 9.2 µL of Nuclease-Free Water (Thermo Fisher Scientific, cat. no. AM9932), 1 µL of 4× SYBR Green (Thermo Fisher Scientific, cat. no. S7563), and 1.8 µL of primer pool at 10 µM. We then performed the PCR reaction using a thermocycler (Biometra GmBH) and the following program: (i) 98 °C for 30 s, (ii) 40 cycles of 98 °C for 15 s and 65 °C for 5 min, (iii) hold at 4 °C. We then pooled an equal volume (20 µL) from each of the six amplicon pools into a 1.5 mL tube and purified DNA with a 1.0 vol/vol ratio of Ampure XP (Beckman Coulter, cat. no. A63881) beads and eluted the purified PCR in 80 µL of Nuclease-Free Water. To measure the DNA concentration in the sample, we used the Qubit dsDNA BR kit (Thermo Fisher Scientific, cat. no. Q32850) according to the manufacturer's instructions.

For multiplexed PCR using the ARTIC (v3) approach, we first diluted the IDT ARTIC nCoV-2019 V3 panel pools 100 µM (IDT, cat. no. 10006788) to a final concentration of 10 µM. To perform the multiplexed PCR, for each of the two primer pools, we aliquoted 6 µL of each cDNA prepared as described above into a PCR mix containing the following reagents: 12.5 µL of NEBNext Q5 Hot Start HiFi PCR Master Mix (NEB, cat. no. M0543L), 2.9 µL of Nuclease-Free Water (Thermo Fisher Scientific, cat. no. AM9932), and 3.6 µL of primer pool at 10 µM. We then performed the PCR reaction with the following program: (i) 98 °C for 30 s, (ii) 35 cycles of 98 °C for 15 s and 63 °C for 5 min, (iii) hold at 4 °C. We then pooled an equal volume (20 µL) from each of the two amplicon pools and purified the DNA with a 0.8 vol/vol ratio of Ampure XP beads and eluted the purified PCR in 40 µL of Nuclease-Free Water. To measure the DNA concentration in the sample, we used the Qubit dsDNA BR kit (Thermo Fisher Scientific, cat. no. Q32850) according to the manufacturer's instructions.

**COVseq**. A detailed step-by-step COVseq protocol is available in the Supplementary Information and at Protocol Exchange[22]. Below, we briefly describe two COVseq workflows, depending on the number of samples to be processed.

*Workflow I (suitable for <10 samples)*. To test the feasibility of COVseq, we initially applied the standard CUTseq protocol to few RNA samples prepared as described

above and processed in individual 0.5 mL tubes. Briefly, we mixed 7 µL (300 ng) of purified pooled PCR product with 1 µL of NlaIII (NEB, cat. no. R0125L), 1 µL of MseI (NEB, cat.no. R0525L), and 1 µL of 10× CutSmart Buffer (NEB, cat. no. B7204S) and incubated the sample for 3 h at 37 °C followed by inactivation for 20 min at 65 °C. Afterwards, we added the following reagents to the same sample (without purifying it) to reach a final volume of 30 µL: 1 µL of NlaIII and 1 µL of MseI adapters (both at 0.33 µM and prepared as we previously described[21]), 1 µL of T4 ligase (Thermo Fisher Scientific, cat. no. EL0011), 3 µL of T4 ligase buffer 10× (Thermo Fisher Scientific, cat. no. EL0011), 2.4 µL of ATP 10 mM (Thermo Fisher Scientific, cat. no. R0441), 0.6 µL BSA 50 mg/ml (Thermo Fisher Scientific, cat. no. AM2616), and 11 µL of Nuclease-Free Water. We incubated the sample for 16 h at 16 °C followed by inactivation for 10 min at 65 °C on the next day. We purified the sample with a 1.2 vol/vol ratio of Ampure XP beads and eluted the purified DNA in 10 µL of Nuclease-Free Water. We performed in vitro transcription (IVT) with the MEGAscript T7 Transcription kit (Thermo Fisher Scientific, cat. no. AM1334) using 8 µL of purified DNA in a final volume of 20 µL and incubated the reaction for 14 h at 37 °C. After IVT, we purified the amplified RNA with a 1.8 vol/vol ratio of RNAClean XP (Beckman Coulter, cat. no. A63987) beads and eluted the purified RNA in 10 µL of Nuclease-Free Water. We then ligated the RA3 adapters by preheating 1 µL of RA3 adapter at 10 µM for 2 min at 70 °C, followed by the addition of 7.8 µL of purified RNA, 1 µL of T4 RNA Ligase 2 truncated (Thermo Fisher, cat. no. M0242L), 1 µL of RNase OUT (Thermo Fisher Scientific, 10777-019), and 1.2 µL of RNA ligase buffer 10× (Thermo Fisher Scientific, cat. no. M0242L) and incubating the mix for 2 h at 25 °C. To reverse transcribe the RNA, we added to the same samples 2 µL of the RT primer at 10 µM pre-heated for 2 min at 70 °C, 2 µL of SuperScript IV reverse transcriptase (Thermo Fisher Scientific, cat. no. 18090050), 5 µL of SuperScript IV buffer 5× (Thermo Fisher Scientific, cat. no. 18090050), 1 µL of 25 mM dNTPs (Thermo Fisher Scientific, cat. no. R1121), 2 µL 0.1 M DTT (Thermo Fisher Scientific, cat. no. 18090050), and 1 µL of RNase OUT (Thermo Fisher Scientific, cat. no. 10777-019) to reach a final volume of 25 µL, and incubated the mix for 20 min at 50 °C followed by an inactivation step of 10 min at 80 °C. After RT, we prepared a PCR mix containing 25 µL of cDNA, 16 µL of RP1 primer, 16 µL of the index primer RPI at 10 µM, 200 µL of NEBNext Ultra II Q5® Master Mix 5× (NEB, cat. no. M0544S), and 143 µL of Nuclease-Free Water. We split the PCR mix into eight strips (50 µL each) and performed PCR in a thermocycler (Biometra GmBH) with the following program: (i) 98 °C for 30 s; (ii) 10 cycles of 98 °C for 10 s, 60 °C for 30 s, 65 °C for 45 s; (iii) 65 °C for 5 min; (iv) hold at 4 °C. We purified the final library with a 0.8 vol/vol ratio of Ampure XP beads and eluted the purified library in 30 µL of Nuclease-Free Water. We measured the DNA concentration in the library using the Qubit dsDNA HS kit (Thermo Fisher Scientific, cat. no. Q32851) and analyzed the fragment size distribution on a Bioanalyzer 2100 (Agilent Technologies, cat. no. G2943CA) using the High Sensitivity DNA kit (Agilent Technologies, cat. no. 5067–4626).

*Workflow II (for >10 samples)*. To process multiple samples in parallel, we performed all reactions until IVT in 384-well plates, leveraging on the I-DOT One nanodispensing device (Dispendix GmbH), which we previously deployed for high-throughput CUTseq[21], to reduce the volume of each reagent and therefore the cost per sample. However, since our I-DOT machine could not be placed inside a BSL-2 lab—which is required to safely handle potentially infectious RNA samples—we used it only for the CUTseq step, while the RT and multiplexed PCR steps were done in a BSL-2 lab using standard multichannel pipettes. In principle, however, all the steps could be implemented on I-DOT, provided that the machine can be placed inside a BSL-2 lab. Briefly, after having manually prefilled each well of a 384-well plate with 5 µL of mineral oil (Sigma-Aldrich, cat. no. M5904) to prevent evaporation during the following steps, we dispensed from 10 to 50 nL of purified pooled PCR amplicons, depending on the Ct values of the samples and then brought up each well to 350 nL with Nuclease-Free Water. After dispensing for each step, we briefly vortexed the plate on a thermomixer (Eppendorf) at 1000 rpm for 1 min and then centrifuged the plate at 3220 × g for 5 min before each incubation. For digestion, we dispensed 150 nL per well of a digestion mix containing 50 nL of NlaIII, 50 nL of MseI, and 50 nL of 10× CutSmart Buffer and incubated the plates at 37 °C for 1 h followed by 65 °C for 20 min to inactivate the enzymes. After digestion, we dispensed 150 nL of NlaIII adapters and 150 nL of MseI adapters (each at 33 nM and prepared as previously described[21]) into each well, followed by 700 nL of a ligation mix containing 200 nL of T4 rapid DNA ligase (Thermo Fisher, cat. no. K1423) or 150 nL of T4 standard DNA ligase (Thermo Fisher Scientific, cat. no. EL0011), 300 nL of T4 ligase buffer (Thermo Fisher, cat. no. K1423) or 150 nL of 10× T4 ligase buffer (Thermo Fisher Scientific, cat. no. EL0011), 120 nL of ATP 10 mM, 30 nL of BSA 50 mg/mL, and 50 nL of Nuclease-Free Water when using rapid ligase or 250 nL when using standard ligase. We incubated the plates at 22 °C for 30 min when using rapid ligase or 1 h for standard ligase, followed by inactivation at 70 °C for 5 min, after which we manually dispensed 5 µL of Nuclease-Free Water/33 nM EDTA (for a final concentration of 25 nM) into each well. We then pooled the contents of multiple wells in the same plate manually or by centrifuging the plate upside down at 117 × g for 1 min, which forces the contents into a collection plate placed at the bottom, and transferred the solution to a 1.5-mL tube. Last, we purified the pooled samples with a 1.2 vol/vol ratio of Ampure XP beads and eluted each pool in 10 µL of Nuclease-Free Water. We prepared sequencing libraries in the same way as described

above for COVseq in single tubes. The number of samples per library depends on the number of differently barcoded adapters available. We have designed 384 different NlaIII and MseI adapters (see Supplementary Data 2), allowing a maximum of 384 samples to be pooled into the same library. However, higher multiplexing could be easily achieved by using a larger number of sequence barcodes when designing COVseq adapters.

**Preparation of SARS-CoV-2 sequencing libraries using a standard approach**. To validate COVseq, we generated individual libraries from 29 left-over samples (OAS-29 samples in Supplementary Data 4) using the NEBNext Ultra II FS DNA Library Prep Kit (NEB, cat. no. E7805L) following the manufacturer's instructions. Briefly, we used 250 ng of the pooled purified amplicons from each sample as input to prepare a library. First, we enzymatically fragmented the amplicons for 7 min at 37 °C followed by incubation for 30 min at 65 °C to achieve a target size around 200 bp. After fragmentation, we performed end-repair and adapter ligation in the same tube followed by purification of the fragments using a 0.8 vol/vol ratio of Ampure XP beads, following the manufacturer's instructions. We amplified adapter-ligated DNA fragments by three PCR cycles with barcoded primers (NEB, cat. no. E7500S) following the manufacturer's instructions and purified the PCR product with a 0.9 vol/vol ratio of Ampure XP beads. We assessed the size distribution and concentration of the libraries on a Bioanalyzer 2100 (Agilent Technologies, cat. no. G2943CA) using the High Sensitivity DNA kit (Agilent Technologies, cat. no. 5067–4626).

**In silico coverage prediction**. We extracted all the cut sites from the SARS-CoV-2 reference genome (NC_045512.2) using a custom Python script. Following this, we predicted the COVseq breadth of coverage by extending known cut site locations in the SARS-CoV-2 genome by the effective read length (theoretical sequence read length minus 20 bp of adapter sequence) on both sides.

**Sequencing**. We sequenced all the NEBNext libraries on the NextSeq 500 system from Illumina using the High Output Kit v2.5 (75 Cycles) (Illumina, Cat. No. 20024906). For COVseq libraries, we used the following kits and platforms: (i) Supernatant sample: High Output Kit v2.5 150 Cycles (Illumina, Cat. No. 20024907) on NextSeq; (ii) OAS-29 samples: High Output Kit v2.5 300 Cycles (Illumina, Cat. No. 20024908) on NextSeq and Reagent Kit v3 600 Cycles (Illumina, Cat. No. MS-102-3003) on MiSeq; (iii) OAS-95 samples: High Output Kit v2.5 300 Cycles (Illumina, Cat. No. 20024908) on NextSeq; (iv) CCI-55 samples: High Output Kit v2.5 150 Cycles (Illumina, Cat. No. 20024907) on NextSeq; (v) CCI-95 samples: Mid Output Kit v2.5 300 Cycles (Illumina, Cat. No. 20024905) on NextSeq.

**Sequencing data pre-processing and variant calling**. We demultiplexed raw sequence reads to fastq files based on index sequences using the BaseSpace Sequence Hub cloud service of Illumina. We then further demultiplexed individual libraries to fastq files for each sample using a custom Python script. Following this, we processed the samples using a Nextflow[24] (version 20.10.0) based analysis pipeline from nf-core[25] called viralrecon[26] (version 1.1.0). In short, we trimmed the adapters from the fastq reads using fastp[27] (version 0.20.1) and aligned them to the SARS-CoV-2 reference genome (NC_045512.2) using bowtie 2[28] (version 3.5.1). Following this, we sorted and indexed the reads using samtools[29] (version 1.9), we trimmed amplicon primer sequences using ivar[30] (version 1.2.2), called variants, and generated the subsequent consensus sequence also using ivar. To determine the percentage of reads that mapped to different organisms and common contaminants, we used FastQ-screen[31] (version 0.14.1). Briefly, 100,000 reads were sampled from the fastq files and aligned to 14 reference sequences using bowtie 2[28] (version 3.5.1) (see Supplementary Table 2). We performed all subsequent analyses using custom R scripts.

**Phylogenetic analyses**. We downloaded sequences and sequence metadata from GISAID[2] (https://www.gisaid.org/ 2021-03-31) and added all the COVseq libraries and relevant metadata from the OAS-29, CCI-55, and OAS-95 samples. We then used the ncov tool (https://github.com/nextstrain/ncov) built on nextstrain[23] to generate a temporal and spatial phylogenetic tree. In addition, we randomly sampled 909 samples from around the world available in GISAID. We analyzed and visualized the resulting newick tree in R using ggtree[32] (version 2.2.4).

**Reporting summary**. Further information on research design is available in the Nature Research Reporting Summary linked to this article.

## Data availability
The BAM files used to generate all the plots in the main Figures and Supplementary Figures have been deposited in the European Nucleotide Archive (ENA) and are available at the following link: https://www.ebi.ac.uk/ena/browser/view/PRJEB42601. All reference sequences used in this study are listed in Supplementary Table 2. All the GISAID data used in this study are described in Supplementary Data 7 and are available at https://www.gisaid.org.

## Code availability
All the custom code used for processing COVseq sequencing data and the custom MATLAB code used in the Cost Analysis (see Supplementary Notes) is available at https://github.com/ljwharbers/COVseq and the repository is linked to Zenodo at the following link: https://doi.org/10.5281/zenodo.4776499.

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

## Acknowledgements
We gratefully acknowledge the authors from the Originating Laboratories and the Submitting Laboratories who generated and shared via GISAID the data on which this research is based[2]. We thank Britta Bouwman (Bienko-Crosetto lab) for critically reading the manuscript and for helping during the final revision. We acknowledge support from the National Genomics Infrastructure in Stockholm funded by Science for Life Laboratory, the Knut and Alice Wallenberg Foundation and the Swedish Research Council, and SNIC/Uppsala Multidisciplinary Center for Advanced Computational Science for assistance with massively parallel sequencing and access to the UPPMAX computational infrastructure and the NGS Facility at the Candiolo Cancer Institute for their support with sequencing. This work was supported by funds from the National Natural Science Foundation of China (no. 81972475) and the Chinese Postdoctoral Science Foundation (2019T120593, 2018M630787) to N.Z.; by a PhD fellowship under the funding of Dipartimenti di Eccellenza 2018–2022 (no. D15D18000410001) to E.B.; by funds from the Fondazione Piemontese per la Ricerca sul Cancro (INTEGRAZIONI DIAGNOSTICA IN ONCOLOGIA—INTERONC FPRC 5×1000 MIUR 2017) to A. Sa. and A. So.; by a SciLifeLab/KAW National COVID-19 Research Program project grant, Research Area Viral Sequence Evolution to N.C.; by a grant from the Swedish Foundation for Strategic Research (SSF BD15-0095) to N.C., through which the I-DOT system was purchased; and by private donations for COVID-19 research from Chiesi Pharma AB and Tetra Pak also to N.C.

## Author contributions
Conceptualization: N.C. Samples: A. Sa., A. So., V.G., M.G.M., and S.B. Data curation: L. H., M.G.M., S.B., and F.C. Formal analysis: L.H., M.B., and N.C. Funding acquisition: N. C. Investigation: M.S., N.Z., M.G.M., S.B., T.T.H.N., and E.B. Methodology: M.S., N.Z., and T.T.H.N. Project administration: N.C. Software: L.H. Supervision: N.C. Visualization: L.H. and N.C. Figure preparation and writing: N.C. and M.B. with contributions from all the authors.

## Competing interests
The authors declare no competing interests.
