## [Peer Review File · Nature Communications]

Reviewers' Comments:

Reviewer #1:

Remarks to the Author:

In this article, Simonetti et al describe COVseq - an adaptation of their previously published CUTseq methodology applied to SARS-CoV-2 whole genome sequencing (WGS). The authors demonstrate the use of COVseq on RNA samples from virus culture and patient specimens and compare their new method to a standard illumina sequencing prep (NEBNext) for SARS-CoV-2 WGS.

The main advantage of COVseq over existing methods (eg NEBNext) is that multiple samples are barcoded and pooled into a single library preparation - rather than one library prep per sample - which improves scalability and reduces cost. In fact, the authors suggest that COVseq can reduce the cost-per-sample of SARS-CoV-2 WGS by 1-2 orders of magnitude compared to standard methods. If true, this would be highly useful for SARS-CoV-2 genomic surveillance initiatives around the world. However, if COVseq is going to be an impactful method, the authors need to demonstrate that it is similarly or more effective to existing methods like NEBNext - not just cheaper. Unfortunately, there are some issues in the way it is presented that make it hard to properly compare. If these can be resolved and, in doing so, if COVseq shows equivalent performance to existing methods, then I would certainly be supportive of publication.

MAJOR ISSUES

In Figure 1e-j, the authors compare COVseq to NEBNext and claim the performance is essentially equivalent. The authors compare the methods according to number of reads (e,f), fraction of SARS-CoV-2 reads (g) and number of SNVs detected (h-i). However, comparisons of coverage breadth, depth and evenness across the SARS-CoV-2 genome are not provided. I would like to see a plot that shows the relationship between between viral titre (Ct values) and coverage breadth (fraction of the genome reaching minimum 10-fold sequencing coverage) for COVseq and NEBNext. I suggest duplicating Fig 1g, but with % genome coverage on the y axis rather than % SARS-CoV-2 reads. Perhaps include comparisons at 1x, 5x and 10x minimum coverage, as in Fig 1c,d. If COVseq does not achieve similar coverage breadth at matched CT values to NEBNext (particularly in the high CT range), then I would be reluctant to adopt it.

Another question on Fig 1e-j: how was overall sequencing depth controlled for? Were the same number of sequencing reads generated for the 30 x NEBNext libraries as the 30 x COVseq libraries, or was set sequenced more deeply than the other? Was it one lane of sequencing each? In panel F the slope of the line suggests the NEBNext samples may be sequenced more deeply but it's a bit hard to tell. Please clarify.

In Fig 1h-i COVseq WF#1 is compared to NEBNext in samples with CT < 35 but in Fig 1j COVseq WF#1 is compared to COVseq WF#2 in samples with CT < 30 (WF#2 is never compared directly to NEBNext). Why did the authors choose two different CT thresholds for these comparisons? Why are WF#1 and WF#2 not compared at CT < 35? If it is because WF#2 performed poorly among samples with CT=30-35 then this is a major limitation of WF#2 that needs to be acknowledged.

The authors compare different replicates of COVseq WF#2 and find that 81.5% of all SNVs identified in samples with CT < 30 are detected in all three replicates. They state that this confirms the reproducibility of the assay but I'm not sure I agree. If I understand correctly, this result means that 20% of detected SNVs are discordant between replicates (ie found in one replicate but not the other). That seems like quite poor reproducibility to me. I would like to see a comparative assessment of reproducibility for NEBNext - does this approach also achieve only 80% reproducibility across 3 replicates? And I would also like to see the comparison performed for samples with CT = 30-35, not just CT < 30. does NEBNext have superior reproducibility for low-titre samples?

The \$20 per sample cost calculation seems to be based on the assumption of sequencing staggeringly large numbers of samples (>10,000) and scale on the modelling curves in Fig 2d,e is such that it is hard to compare the cost of the different methods at more modest sample numbers eg 10-1000 samples. Could the authors reproduce these plots to give us an idea how the costs compare in this lower range of samples numbers, which is probably more realistic for most labs.

SECONDARY ISSUES / SUGGESTIONS

The authors only compare their method to standard illumina library preps (NEBNext, Nextera etc). However, several other methods for massively parallel SARS-CoV-2 sequencing have been developed. For example SwabSeq from the company Octant:

<https://www.ncbi.nlm.nih.gov/pmc/articles/PMC7480060/>

<https://www.octant.bio/swabseq>

I realise it may not be possible to make a direct comparison, but it would be good if the authors could mention SwabSeq and other similar methods and discuss the pros and cons of each. Is there any reason to believe COVseq is superior to SwabSeq etc?

I realise things are moving at a rapid pace, but it would be good to mention all of the recently emerging SARS-CoV-2 variants of concern (eg B.1.1.7, B.1.351, B.1.346, P.1) and whether these would all be effectively detected by COVseq. At the very least this will help increase the level of interest in the article.

The authors show increasing single-end read length (SE75/150/300) leads to increase coverage breadth in COVseq. I wonder whether it would also be possible to pair COVseq with an Oxford Nanopore library prep to increase read length and achieve better coverage depth?

Reviewer #2:

Remarks to the Author:

Review comments

The authors present COVseq, an adaptation of the CUT-seq technique to sequence viral genomes with enhanced cost-effectiveness. In the manuscript, the authors show both the performance of COVseq to sequence viral genome and also the reduced cost thanks to the multiplexed protocol. Given that it is very important to sequence as many viral genomes as possible to keep on track the viral evolution of SARS-COV-2, it is a very relevant study to fight against the pandemic situation.

I have a few points and suggestions.

1. It is shame that there are "dark genomes" that are not currently reachable by the current set of restriction enzymes. While it is theoretically possible to cover that region with another restriction enzyme, what would be then the consequence of that? (I presume that there would incur extra

cost).

2. It appears that the most of genome is covered (x1, x5, x10 reads) by the current restriction enzyme already. I am curious why there are still 69 SNVs not detected by the COVseq technique (Fig. 1i). Are these all within dark regions with a low amount of reads? Or is there any possibility that they are FP?

3. I wonder what is the minimum number of reads required to call variants. Maybe authors check the number of reads on the missed variants by COVseq. There might be a filtering step in the variant detection pipeline as well.

4. I think it is informative to show the absolute number of reads per variant instead of the number of reads per million, as it is now in Fig 2A. In particular, since this is a multiplexed approach, I am curious about the number of reads per sample (and variation across the samples). Can authors also measure the fraction of the genome with sufficient coverage per sample?

5. In Figure 2 b-c, is it expected that COVseq samples form a distinct family? Is there any explanation, for instance, the pandemic phase and/or region in which COVseq RNA is obtained?

6. There is no red dashed line in Fig 1c,d - unlike what is stated in the Legend.

Point-by-point response to the Reviewers' comments

We would like to thank both Reviewers for their constructive comments, which greatly helped us strengthen our manuscript and present the COVseq method in a clearer manner.

Following the Reviewers' suggestions, we have done all the requested revisions and further expanded the number of samples analyzed and depth of analyses. In particular:

- In our continuous strive to improve and optimize COVseq, we have determined that the COVseq workflow that includes a PCR purification step before the sample barcoding (workflow #1 in our original manuscript) provides the highest quality of results. In addition, we have added an inactivation step after the ligation of barcodes, which eliminates the risk of sample cross talk before IVT. We now consider this as standard COVseq workflow and, accordingly, we have prepared new libraries from the 30 samples originally described in Figure 1 using this optimized workflow. This new workflow coupled with longer read sequencing (PE150 on NextSeq 500 and PE300 on MiSeq) allowed us to achieve >98% of SARS-CoV-2 genome coverage at 10X depth and, consequently, a much higher overlap of called SNVs between COVseq and the commercial NEBNext method, which we used as benchmark (98.1% overlap out of 424 SNVs detected in 29 samples sequenced by both COVseq and NEBNext, see **new Fig. 1j**). NOTE: of the 30 samples originally described in Fig. 1, only 29 samples had a sufficient amount of left-over RNA for preparing new libraries. That is why, in the revised manuscript, this cohort now comprises only 29 samples (OAS-29 samples, see new nomenclature in the **revised Supplementary Table 4**).
- We have further expanded the cohort of samples sequenced by COVseq, by applying the aforementioned newly optimized COVseq workflow to a new set of 95 samples (see samples OAS-95 in the **revised Supplementary Table 4**) from which we prepared three replicate COVseq libraries—including 7 cases of UK variant (B.1.1.7) and 87 cases from a nosocomial Covid-19 outbreak that occurred in Jan 2021 at a hospital in Turin, Italy. The results of this new experiment—which further highlight the reproducibility of COVseq—are presented in the **revised Figure 2**, which has been entirely modified to showcase the reproducibility of COVseq and its ability to identify the UK variant currently dominating in Italy and other EU countries.
- Using our newly optimized COVseq workflow, we have also re-prepared libraries, (each in four replicates) from all the 55 samples, which we originally described in our manuscript and re-sequenced them using longer reads (PE150 on NextSeq 500), in order to achieve higher genome coverage. The results of these new experiments, which now show a much higher degree of overlap between replicates compared to the results shown before (91.4% of all SNVs detected in at least three replicates) and further corroborate the reproducibility of COVseq, are presented in the **revised Supplementary Fig. 4**.
- We have repeated the phylogenetic analysis originally shown in Figure 2, by including all the newly sequenced samples and expanding the number of GISAID sequences used for clustering. The results of these analyses are now presented separately in the **newly added Fig. 3**. In particular, the **new Fig. 3c** demonstrates the applicability of COVseq in cluster analysis, by showing the ability of our method to resolve phylogenetic relationships between the newly sequenced samples from the nosocomial outbreak mentioned above.

- For clarity and to avoid downsizing the phylogenetic trees in the new Fig. 3, we have now moved the results of our cost analysis to a **newly added Fig. 4**. Since we have now identified an optimal COVseq workflow, we have removed the original distinction between three workflows, and accordingly only show this optimized COVseq workflow in our cost analysis.
- To further streamline the COVseq workflow, we have implemented a different SARS-CoV-2 genome amplification step, by adopting the multiplex PCR protocol (V3) developed by the ARTIC network (see **new Supplementary Fig. 5a**). Although due to time constraints we could not repeat all the original experiments (including the comparison with NEBNext) using this more streamlined workflow, we are now testing this approach in the frame of a SARS-CoV-2 genomic surveillance program, which we recently started in the Piemonte Region in Italy. The **new Fig. 4a-c**, which is based on our updated cost analysis, shows how ARTIC-COVseq further increases the cost-effectiveness of COVseq compared to three different commercial kits. In the **new Supplementary Fig. 5b**, we show an example of variants that we have recently been able to detect using ARTIC-COVseq within the aforementioned genomic surveillance program, including 19 cases that were confirmed using an orthogonal commercial kit (CleanPlex).

We hope that the Reviewers will appreciate our efforts to further improve our method and find our revised manuscript now suitable for *Nature Communications*. We reply to individual Reviewers' questions in line below.

Reviewer #1

In this article, Simonetti et al describe COVseq - an adaptation of their previously published CUTseq methodology applied to SARS-CoV-2 whole genome sequencing (WGS). The authors demonstrate the use of COVseq on RNA samples from virus culture and patient specimens and compare their new method to a standard illumina sequencing prep (NEBNext) for SARS-CoV-2 WGS.

The main advantage of COVseq over existing methods (eg NEBNext) is that multiple samples are barcoded and pooled into a single library preparation - rather than one library prep per sample - which improves scalability and reduces cost. In fact, the authors suggest that COVseq can reduce the cost-per-sample of SARS-CoV-2 WGS by 1-2 orders of magnitude compared to standard methods. If true, this would be highly useful for SARS-CoV-2 genomic surveillance initiatives around the world. However, if COVseq is going to be an impactful method, the authors need to demonstrate that it is similarly or more effective to existing methods like NEBNext - not just cheaper. Unfortunately, there are some issues in the way it is presented that make it hard to properly compare. If these can be resolved and, in doing so, if COVseq shows equivalent performance to existing methods, then I would certainly be supportive of publication.

We thank the Reviewer for appreciating our effort to demonstrate the analytical validity of COVseq and for her/his constructive comments.

MAJOR ISSUES

In Figure 1e-j, the authors compare COVseq to NEBNext and claim the performance is essentially equivalent. The authors compare the methods according to number of reads (e,f), fraction of SARS-CoV-2 reads (g) and number of SNVs detected (h-i). However, comparisons of coverage breadth, depth and evenness across the SARS-CoV-2 genome are not provided. I would like to see a plot that shows the relationship between viral titre

(Ct values) and coverage breadth (fraction of the genome reaching minimum 10-fold sequencing coverage) for COVseq and NEBNext. I suggest duplicating Fig 1g, but with % genome coverage on the y axis rather than % SARS-CoV-2 reads. Perhaps include comparisons at 1x, 5x and 10x minimum coverage, as in Fig1c,d. If COVseq does not achieve similar coverage breadth at matched CT values to NEBNext (particularly in the high CT range), then I would be reluctant to adopt it.

We thank the Reviewer for raising these important issues and for the suggestions. As mentioned above, we have now further optimized the COVseq workflow and repeated all the previous experiments using this workflow and longer sequencing reads, which allowed us to achieve a much higher overlap in the number and type of SNVs identified by NEBNext and COVseq, as shown in the **new Fig. 1i and j**. Following the reviewer's suggestion, in the **new Fig. 1g** we now show that the breadth of coverage at 10X is highly correlated between COVseq and NEBNext (Pearson's correlation coefficient, PCC: 0.95) and that COVseq achieves a slightly higher breadth of coverage at high Ct value samples in comparison to NEBNext. Furthermore, in the **new Supplementary Fig. 2b and c** we now show that the 10X breadth of coverage and number of SNVs per sample is essentially identical when COVseq libraries are sequenced on two different platforms (PE150 on NextSeq 500 and PE300 on MiSeq).

Another question on Fig 1e-j: how was overall sequencing depth controlled for? Were the same number of sequencing reads generated for the 30 x NEBNext libraries as the 30 x COVseq libraries, or was set sequenced more deeply than the other? Was it one lane of sequencing each? In panel F the slope of the line suggests the NEBNext samples may be sequenced more deeply but it's a bit hard to tell. Please clarify.

In the original Fig. 1f, NEBNext libraries were indeed sequenced deeper than COVseq libraries, however this did not affect our ability to perform reliable SNV calling in the latter case. As discussed above, we have now re-sequenced 29 of the 30 samples originally used to compare COVseq and NEBNext (now labeled OAS-29 samples, see **revised Supplementary Table 4**): the overall sequencing depth in COVseq is again lower than NEBNext, due to the fact that the samples were sequenced this time on MiSeq instead of NextSeq 500 as before, to increase the sequencing read length (see summary of sequencing results in the **revised Supplementary Table 3**). However, these results indicate that, even at a lower sequencing depth, COVseq is able to achieve similar results as NEBNext. We would like to emphasize that, independently of the sequencing platform used, it would be difficult to achieve an equal amount of reads for NEBNext and COVseq, since NEBNext libraries are generated from individual samples, whereas COVseq libraries contain multiple samples that are pooled together before the *in vitro* transcription step in the COVseq protocol.

In Fig 1h-i COVseq WF#1 is compared to NEBNext in samples with CT < 35 but in Fig 1j COVseq WF#1 is compared to COVseq WF#2 in samples with CT < 30 (WF#2 is never compared directly to NEBNext). Why did the authors choose two different CT thresholds for these comparisons? Why are WF#1 and WF#2 not compared at CT < 35? If it is because WF#2 performed poorly among samples with CT=30-35 then this is a major limitation of WF#2 that needs to be acknowledged.

We apologize to the Reviewer for this confusion. We have now removed this comparison from Fig. 1, as we do not distinguish any more between different COVseq workflows. In the **new Fig. 1i and j**, we now compare SNVs between COVseq and NEBNext using 20 samples with Ct value ≤ 35 , for which the breadth of coverage at 10X was sufficiently high to secure reliable SNV calling.

The authors compare different replicates of COVseq WF#2 and find that 81.5% of all SNVs identified in samples with CT < 30 are detected in all three replicates. They state that this confirms the reproducibility of the assay but I'm not sure I agree. If I understand correctly, this result means that 20% of detected SNVs are discordant between replicates (ie found in one replicate but not the other). That seems like quite poor reproducibility to me. I would like to see a comparative assessment of reproducibility for NEBNext - does this approach also achieve only 80% reproducibility across 3 replicates? And I would also like to see the comparison performed for samples with CT = 30-35, not just CT < 30. does NEBNext have superior reproducibility for low-titre samples?

As mentioned above, we have now repeated this experiment, by preparing four new replicate libraries using the same 55 samples and sequencing them with longer reads (PE150 vs. SE150 in the original manuscript). In addition, we have prepared three replicate COVseq libraries from an entirely new set of 95 samples (OAS-95 samples in the **revised Supplementary Table 4**). As shown in the **entirely new Fig. 2 and new Supplementary Fig. 3 and 4**, the reproducibility of COVseq, both in terms of the breadth of coverage at 10X sequencing depth and of number of SNVs called in each sample, is extremely high, making us confident that our method is very reliable. We therefore think that it would be of little added value to assess the reproducibility of NEBNext, especially considering the high degree of concordance that we now show between the two methods.

The \$20 per sample cost calculation seems to be based on the assumption of sequencing staggeringly large numbers of samples (>10,000) and scale on the modelling curves in Fig 2d,e is such that it is hard to compare the cost of the different methods at more modest sample numbers eg 10-1000 samples. Could the authors reproduce these plots to give us an idea how the costs compare in this lower range of samples numbers, which is probably more realistic for most labs.

We thank the Reviewer for raising this point and for the suggestion. Accordingly, in the **new Fig. 4** we now show a zoom-in view of the cumulative cost curves for the first 1,000 samples processed and discuss this in depth in the **revised Supplementary Notes**. The Reviewer is right that, in the low-sample-number regime, COVseq might not be competitive with existing commercial solutions. However, for centralized laboratories having to process a hundred or potentially even thousands of samples, COVseq would be clearly a more cost-effective solution. In this context, we do believe that COVseq would be particularly beneficial for low-income countries, where only centralized laboratories (e.g., national health agencies) can realistically perform genomic surveillance. Notably, these are also the countries in which genomic surveillance is crucially needed, since new variants of concern can emerge as a consequence of the very slow vaccination rates in these countries.

SECONDARY ISSUES / SUGGESTIONS

The authors only compare their method to standard illumina library preps (NEBNext, Nextera etc). However, several other methods for massively parallel SARS-CoV-2 sequencing have been developed. For example SwabSeq from the company Octant:

<https://www.ncbi.nlm.nih.gov/pmc/articles/PMC7480060/>

<https://www.octant.bio/swabseq>

I realise it may not be possible to make a direct comparison, but it would be good if the authors could mention SwabSeq and other similar methods and discuss the pros and cons of each. Is there any reason to believe COVseq is superior to SwabSeq etc?

We thank the Reviewer for this valuable suggestion. However, we note that SwabSeq is designed for high-throughput SARS-CoV-2 detection by sequencing, and not for WGS of the viral genome. Therefore, we believe that SwabSeq and COVseq represent

complementary, rather than competing methods in the management of the ongoing pandemic. We now cite SwabSeq in the Introduction as well as in the Discussion in our revised manuscript.

I realise things are moving at a rapid pace, but it would be good to mention all of the recently emerging SARS-CoV-2 variants of concern (eg B.1.1.7, B.1.351, B.1.346, P.1) and whether these would all be effectively detected by COVseq. At the very least this will help increase the level of interest in the article.

We thank the Reviewer for this suggestion. Accordingly, in the **new Fig. 1k** we now show that all the SNVs defining the B.1.1.7, B.1.351, and P.1 variants can potentially be detected by COVseq, and in the **new Fig. 2e** we actually show the ability of COVseq to correctly identify samples belonging to the B.1.1.7 (UK) variant. Furthermore, as mentioned above, we are now running a genomic surveillance program based on COVseq, which shows the ability of our method to also capture other lineages reported thus far, as exemplified in the new **Supplementary Fig. 5b**.

The authors show increasing single-end read length (SE75/150/300) leads to increase coverage breadth in COVseq. I wonder whether it would also be possible to pair COVseq with an Oxford Nanopore library prep to increase read length and achieve better coverage depth?

We thank the Reviewer for this suggestion. In principle, we do not see any obstacle in adapting the COVseq workflow to the Nanopore platform. However, since we do not have immediate access to this platform in our lab, we have not been able to perform a proof-of-principle experiment demonstrating the feasibility of COVseq on Nanopore.

Reviewer #2

The authors present COVseq, an adaptation of the CUT-seq technique to sequence viral genomes with enhanced cost-effectiveness. In the manuscript, the authors show both the performance of COVseq to sequence viral genome and also the reduced cost thanks to the multiplexed protocol. Given that it is very important to sequence as many viral genomes as possible to keep on track the viral evolution of SARS-COV-2, it is a very relevant study to fight against the pandemic situation.

We thank the Reviewer for appreciating our work and recognizing the potential utility of our method in fighting the ongoing pandemic.

I have a few points and suggestions.

1. It is shame that there are "dark genomes" that are not currently reachable by the current set of restriction enzymes. While it is theoretically possible to cover that region with another restriction enzyme, what would be then the consequence of that? (I presume that there would incur extra cost).

We thank the Reviewer for raising this important point. As mentioned above, we have now further optimized our COVseq workflow and prepared new libraries from all the previously described 85 samples except one (for which no RNA was left over) and sequenced them with longer sequencing reads (PE150 on NextSeq or PE300 on MiSeq). As a result, in the **revised Fig.1** we now report a much higher breadth of coverage at 10X sequencing depth than we did before, which in turn resulted in essentially complete overlap in the number of SNVs called by COVseq and the NEBNext benchmark. In addition, as mentioned above, we have now further expanded the number of samples sequenced, by processing in triplicate 95 new samples, including 7 cases of suspected UK variant (see the entirely **revised Fig. 2**). PE150 sequencing of these samples on NextSeq resulted in >98% SARS-CoV-2 genome coverage at 10X (see **new Fig. 2a and 2b**) and in turn in very high reproducibility in the number and type of SNVs identified (see **new Fig. 2c-e**). Therefore, we are confident that, with our newly optimized workflow coupled with longer read sequencing, COVseq can provide full SARS-CoV-2 genome coverage without the need to use a third restriction enzyme, as we initially proposed.

We did, however, explore whether adding a third restriction enzyme (Bfal) would be feasible and result in higher coverage, as we expect theoretically. Unfortunately, so far we have not been able to generate COVseq libraries using the combination of three enzymes (Bfal, MseI and NlaIII), which we previously proposed, probably because this results in too short fragments. It is possible that using different combinations of enzyme concentration and/or incubation time we would eventually manage to obtain sequenceable libraries. However, given the urgency imposed by the ongoing pandemic and considering that by using longer reads we have been able to solve the problem of 'dark regions', which we previously reported, we have decided for now to use a combination of two enzymes only. Accordingly, in the revised manuscript we have removed all the text and related figures about 'dark regions' and only show data obtained with MseI and NlaIII.

2. It appears that the most of genome is covered (x1, x5, x10 reads) by the current restriction enzyme already. I am curious why there are still 69 SNVs not detected by the COVseq technique (Fig. 1i). Are these all within dark regions with a low amount of reads? Or is there any possibility that they are FP?

As explained above and also in the response to Reviewer 1, we have now further optimized our COVseq protocol and used longer sequencing reads, which resulted in much higher overlap between COVseq and NEBNext (see **new Fig. 1i and j**).

3. I wonder what is the minimum number of reads required to call variants. Maybe authors check the number of reads on the missed variants by COVseq. There might be a filtering step in the variant detection pipeline as well.

We thank the Reviewer for this suggestion to further optimize variant calling. We currently only retain called SNVs if they pass all quality controls, one of them being a minimum read depth of 15 at the variant location. It is indeed the case that variants that are missed by COVseq, but are detected by NEBNext, have a low read depth in NEBNext and COVseq, as we show in the **new Supplementary Fig. 2d**. NEBNext was able to detect these samples because of the higher sequencing depth of NEBNext libraries.

4. I think it is informative to show the absolute number of reads per variant instead of the number of reads per million, as it is now in Fig 2A. In particular, since this is a multiplexed approach, I am curious about the number of reads per sample (and variation across the samples). Can authors also measure the fraction of the genome with sufficient coverage per sample?

We thank the Reviewer for raising this point. The level of variability in number of reads between samples can be seen in **revised Fig. 1e-f** and **new Supplementary Fig. 2a**. Additionally, we have used a breadth of coverage at a read depth of 10X as a measurement of sufficient coverage as shown in the **revised Fig. 1g**, **new Supplementary Fig. 2b**, and **new Supplementary Fig. 3a-c**.

5. In Figure 2 b-c, is it expected that COVseq samples form a distinct family? Is there any explanation, for instance, the pandemic phase and/or region in which COVseq RNA is obtained?

To do:

We have now repeated this analysis including 95 extra samples (OAS-95 samples in the **revised Supplementary Table 4**) and show the results in a completely new figure (see **new Fig. 3**). In this phylogenetic analysis and in previous analyses we have seen that samples from our different sample cohorts look highly similar within their respective cohort (**see new Fig. 1j**, **new Fig. 2e** and **new Supplementary Fig. 5b**) This indicates that indeed, pandemic phase and/or region in which the RNA is obtained is likely a strong determinant of the observed viral genome composition.

6. There is no red dashed line in Fig 1c,d - unlike what is stated in the Legend.

We apologize for this mistake and have corrected it.

Reviewers' Comments:

Reviewer #1:

Remarks to the Author:

The authors have done an excellent job of improving both their COVseq workflow and the presentation of the data. I am particularly happy with the inclusion of Fig 1g and Fig 2a, which both suggest the new COVseq workflow is performing very effectively, and perhaps outperforming NEBnext among high-CT samples.

I'm glad the authors no longer compare Workflow 1 vs Workflow 2, as I found that a bit confusing in the previous manuscript version.

Given my concerns have been addressed, I am happy to support publication of the manuscript.

Well done :)

Reviewer #2:

Remarks to the Author:

The authors improved the COV-seq protocol using paired-end sequencing (PE150 and PE300) instead of single-end sequencing, achieving near-complete coverage of 98.8% of the whole genome.

In general, I would agree with the authors that COV-seq is a very attractive approach for tracking the evolution of SARS-COV-2. However, I have a few comments, mostly concerns about how to interpret the results.

1. The authors often rely on the Pearson correlation coefficients (PCC) to compare methods. Though significantly high PCC is impressive, PCC may not reflect the general shift in the signal. Due to this, some performance difference is overlooked in several places. I suggest putting the $y=x$ line instead of (or on top of) the regression line to make the comparison fair.

For instance:

- In Fig 1g, COVseq achieved a higher 10x breadth of coverage than NEBNext.

- In Supplementary Fig 2b. It seems PE300 achieved a higher 10x breadth of coverage than PE150. It is better to show the $y=x$ line instead of the regression line. Current visualization is misleading.

- In Supplementary Fig 2c. The same applies here. NEBNext tend to detect more SNVs than COVseq PE150. Also, the x-axis label needs to fix. (12 and 9 swapped) I am not sure how it is related to Fig 1i, which shows the same performance between COVseq and NEBNext. (is it PE300 in Fig. 1i?)

2. I suspect that, though it is still very attractive, COVseq (at least with PE150) is slightly less sensitive than NEBNext. As I pointed out, NEBNext tend to detect more SNVs than COVseq (Supp Fig 2C). Also, according to Figure 1k, certain regions tend to be covered less well, quite consistently. Therefore, I would suggest the following.

- Yes, 98.8% coverage is very impressive (but this is with PE300). Is 1.2% does cover any of the VOCs? How about variants with dark regions in Fig 1k? Are they among this 1.2% of regions? Or happened to be close by? I think this clarification is still useful.

- Given that the detection of SNVs depends on the number of raw reads (as demonstrated in Supp Fig 2d; where SNVs are missed in the regions with low coverage), I would not use normalized count (in Figure 1k). Furthermore, if there is a minimum read count for SNV detection, I would clearly show if the number of reads exceeds the cutoff or not at each SNV location.

- According to the above points, I would suggest clearly show the dark regions & technological limitations of COV-seq. I believe, though, the major statement of the applicability of COVseq to track SARs-COV-2 evolution still holds.

Point-by-point reply to the Reviewers' comments

Reviewer #1

The authors have done an excellent job of improving both their COVseq workflow and the presentation of the data. I am particularly happy with the inclusion of Fig 1g and Fig 2a, which both suggest the new COVseq workflow is performing very effectively, and perhaps outperforming NEBnext among high-CT samples.

I'm glad the authors no longer compare Workflow 1 vs Workflow 2, as I found that a bit confusing in the previous manuscript version.

Given my concerns have been addressed, I am happy to support publication of the manuscript.

Well done :)

We thank the Reviewer for appreciating our efforts to improve our original manuscript throughout the revision process, and are very pleased that the Reviewer is now supportive of publication of our manuscript in *Nature Communications*.

Reviewer #2

The authors improved the COV-seq protocol using paired-end sequencing (PE150 and PE300) instead of single-end sequencing, achieving near-complete coverage of 98.8% of the whole genome.

In general, I would agree with the authors that COV-seq is a very attractive approach for tracking the evolution of SARS-COV-2. However, I have a few comments, mostly concerns about how to interpret the results.

We thank the Reviewer for appreciating our efforts to improve the COVseq protocol and for recognizing the attractiveness of our method for genomic surveillance of SARS-CoV-2. We also thank the Reviewer for providing further constructive comments and suggestions, which we have now addressed as discussed below.

1. The authors often rely on the Pearson correlation coefficients (PCC) to compare methods. Though significantly high PCC is impressive, PCC may not reflect the general shift in the signal. Due to this, some performance difference is overlooked in several places. I suggest putting the $y=x$ line instead of (or on top of) the regression line to make the comparison fair.

For instance:

- In Fig 1g, COVseq achieved a higher 10x breadth of coverage than NEBNext.

We thank the Reviewer for pointing this out and suggesting using the bisector $y=x$ line instead of the regression line. We have now changed this in Fig. 1g.

- In Supplementary Fig 2b. It seems PE300 achieved a higher 10x breadth of coverage than PE150. It is better to show the $y=x$ line instead of the regression line. Current visualization is misleading.

We now show the $y=x$ line in Supplementary Fig 2b as well.

- In Supplementary Fig 2c. The same applies here. NEBNext tend to detect more SNVs than COVseq PE150. Also, the x-axis label needs to fix. (12 and 9 swapped) I am not sure how it is related to Fig 1i, which shows the same performance between COVseq and NEBNext. (is it PE300 in Fig. 1i?)

We thank the Reviewer for pointing this out, we have now rectified this. Fig. 1i indeed refers to COVseq with PE300 sequencing. We have now clarified this in the corresponding figure legend. Also here, we now show the $y=x$ line.

2. I suspect that, though it is still very attractive, COVseq (at least with PE150) is slightly less sensitive than NEBNext. As I pointed out, NEBNext tend to detect more SNVs than COVseq (Supp Fig 2C). Also, according to Figure 1k, certain regions tend to be covered less well, quite consistently. Therefore, I would suggest the following.

- Yes, 98.8% coverage is very impressive (but this is with PE300). Is 1.2% does cover any of the VOCs? How about variants with dark regions in Fig 1k? Are they among this 1.2% of regions? Or happened to be close by? I think this clarification is still useful.

We thank the Reviewer for this comment. We now show the location of so-called 'dark regions' as well as of SNVs in the UK, South African and Brazilian VOC in the **newly added Supplementary Figure 1a**. Assuming to perform SE300, only 1 out of 32 SNVs in these VOC would fall into one of these dark regions and thus not be covered. However, as shown in the **newly amended Fig. 1k**, PE300 on MiSeq can cover the location of this SNV.

- Given that the detection of SNVs depends on the number of raw reads (as demonstrated in Supp Fig 2d; where SNVs are missed in the regions with low coverage), I would not use normalized count (in Figure 1k). Furthermore, if there is a minimum read count for SNV detection, I would clearly show if the number of reads exceeds the cutoff or not at each SNV location.

We thank the Reviewer for this suggestion. Accordingly, we have now modified Fig. 1k to display raw read counts instead of normalized counts. We have also colored SNV locations that, in our experiment, were covered by less than 15 reads, which is the threshold used in our pipeline for calling SNVs.

- According to the above points, I would suggest clearly show the dark regions & technological limitations of COV-seq. I believe, though, the major statement of the applicability of COVseq to track SARs-COV-2 evolution still holds.

As mentioned above, we now show this in the **newly added Supplementary Fig. 1a**.